# Electromagnetic Soil Characterization for Undergrounded RFID System Implementation

**Ayman Elboushi** [1,*], **Ahmed Telba** [2], **Abdelrazik Sebak** [3] **and Khalid Jamil** [2]

[1] Electronics Research Institute (ERI), Cairo 12622, Egypt
[2] Department of Electrical Engineering, King Saud University, Riyadh 11451, Saudi Arabia; atelba@ksu.edu.sa (A.T.); k.jamil@gmail.com (K.J.)
[3] Department of Computer and Electrical Engineering, Concordia University, Montreal, QC H3G 1M8, Canada; abdo@ece.concordia.ca
[*] Correspondence: a_m_fekry@eri.sci.eg

**Abstract:** This paper can be divided into two main parts. In the first part, an extensive experimental study for electromagnetic characteristic assessment of different soil samples is presented. In the second part of the paper, a practical verification of the obtained link budget model is performed using a buried metal-backed RFID tag antenna under a 40 cm sand layer. This antenna is designed to operate at 915 MHz with MONZA3 chip. Using Impinj Speedway system the tag antenna could be detected, and its information could be read from different distances of up to more than 2.5 m.

**Keywords:** soil EM characterization; electrical permittivity; underground RFID; oil mining; RFID measurement

---

## 1. Introduction

Radio Frequency Identification (RFID) is an automatic short-range and fast wireless data collection, technology with long historical roots [1,2]. Moreover, it has grown significantly in recent years and new applications have appeared. One of the possible benefits of such technology is the possibility to use it for locating any underground-utilities in wide areas where there are not many nearby surface landmarks to use as reference points. In [3], a passive harmonic tag for undergrounded asset localization is presented for utility localization. However, buried object localization, such as pipes, sometimes is not enough, where more information about the nature of the fluid transferred, its viscosity, and its temperature are required. One of the possible solutions for this problem is to use a passive RFID tag antenna with a chip for storing all the required information. The tag antenna is connected to the chip via a differential port. The input impedance of the tag antenna is designed according to the chip type used in the RFID system, which usually has a real and imaginary part ($Z\_chip = X \pm jY$). So, for satisfying maximum power transfer, the antenna input impedance should be a complex conjugate of the chip impedance (i.e., $Z\_antenna = X \mp jY$). The operating frequency of the RFID system can vary according to the nature of the environment. For example, in the underground environment, the most suitable range is UHF (850–950 MHz) [3] to minimize the propagation losses. A lot of researcher efforts all over the world have been exerted to design proper tag antennas [4–7]. whereas, the efficient tag antenna should be characterized by suitable impedance matching with the chip, stable radiation characteristics over the operating band, small size, and low fabrication cost.

However, before talking about the design of a high-performance tag antenna, the soil electromagnetic (EM) characteristics should be studied extensively, where the soil is the medium for EM wave propagation from the reading antenna to the buried tag. In addition, for achieving successful connection between the reader and tag antenna a certain amount of power level should be maintained

(threshold power of the RFID chip). So, a complete link budget analysis based on the soil EM characters becomes mandatory.

Many methods can be employed for soil characterization. A chemical based method is proposed in [8] for electrical resistivity evaluation of a soil sample using $CO_2$ carbonization for different time durations. However, in [9] the electrical resistivity of the soil is evaluated using a portable Time Domain Reflectometer (TDR), where the measurements are acquired in-time by continuous Frequency Domain Reflectometer (FDR) sensors. Moreover, geophysical methods of vertical electrical sounding, four-electrode probe, non-contact electromagnetic profiling, and self-potential were modified for measuring soil electrical properties and tested in different soil studies as presented in both [10], and [11]. An artificial intelligence approach is employed to investigate the correlation between electrical resistivity, obtained using one of the pre-mentioned techniques, and soil-water content in order to better the results from conventional technique systems [12]. In the ITU-R report [13], the complex and real dielectric constants of different soil types (sand, clay, and silt) are presented over the frequency band from 1 MHz to 30 MHz, where the effective parameters can be used with homogeneous smooth earth ground-wave propagation scenarios. In addition, complex dielectric permittivity is measured for two soil samples (silt loam, and silty clay loam), as introduced in [14], using the dielectric relaxation spectroscopy by means of the coaxial Transmission Line method (CTL) over the 1 MHz to 10 GHz frequency range.

In the first part of this paper, thorough analysis of seven soil samples is conducted for soil electrical properties, i.e., relative permittivity, electrical conductivity, and tangential losses. However, in this study, a simpler method based on a ready-made dielectric assessment probe is utilized. The measurement probe, represents a coaxial cavity where, its resonance frequency response can be modified according to the backed material (the material under test). The Vector Network Analyzer (VNA) is utilized to record the frequency response of the measurement probe. A ready-made software package, provided by SPEAG, is used for estimating the various electrical properties of the soil samples.

The second part shows a practical test, carried out on a system level, for a metal backed buried tag antenna. The input impedance of the proposed tag antenna is measured using the method mentioned in [8]. The tag antenna can be localized under 40 cm of yellow sand accurately, and the stored information in the chip can be collected properly.

## 2. Experimental Soil Characterization

Soil characterization models and link budget analysis for different soil samples, taken from different areas, become mandatory to provide an accurate link budget model for undergrounded RFID systems. The Dielectric Assessment Kit (DAK) provided by SPEAG is employed to obtain high-precision dielectric parameter measurements (permittivity, conductivity, and loss tangent).

These soil analysis tests were carried out for seven different soil samples taken from Riyadh city. The picked soil samples included yellow sand, red sand, black soil, gray soil, black gravels, white gravels, and red/orange gravels. About 500 mm$^3$ of each soil type was considered as a test sample. In order to ensure accurate results of the tests, the DAK system was calibrated using both distilled water and copper metal sheets before starting the practical measurements. A photo for the DAK system setup is shown in Figure 1. The DAK probe is pressed against the 500 mm$^3$ soil sample, while it is connected from its end-terminal to the Vector Network Analyzer (VNA) through a 50-ohm coaxial cable. The VNA measures the frequency response of the probe. The measured data are collected and analyzed via a software package provided by SPEAG installed on a PC. Furthermore, the software calculates the various electrical properties of the soil sample over a pre-defined frequency range.

Figures 2–4 show the measured results of the EM characteristics of the soil samples (real part of electrical permittivity, conductivity, and tangential losses, respectively). The measurements were carried out over two frequency bands, the first from 850 to 950 MHz, while the second one from 1 to 5 GHz.

Moreover, the results illustrate the effect of the humidity and the water content effect on the soil properties. Besides dry soil, 5% and 10% water content are considered in the modelling process. It can be

concluded that the increment of the water content in the soil increases the real permittivity, the electrical conductivity, and the tangential loss, which results in increasing the path loss inside the soil.

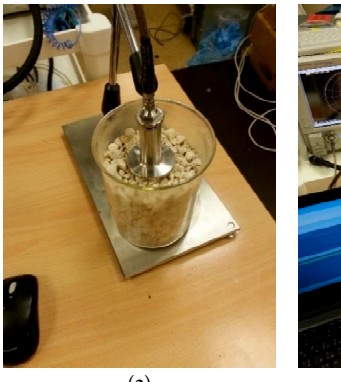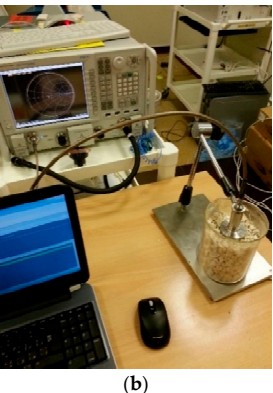

(**a**)          (**b**)

**Figure 1.** The practical setup for soil measurements. (**a**) the sample under test. (**b**) DAK system setup.

Furthermore, it can be seen that the black gravels achieved the lowest relative permittivity compared with other soil samples. It can be accounted to the presence of bigger air gaps and the larger inner spaces between the soil particles. This phenomenon also, has a great impact on the electrical conductivity of the same sample, as it has the lowest conductivity among other samples. The variation of the electrical properties of the soil samples versus the frequency is observed to be smooth or sometimes constant over the lower frequency band of 800 MHz to 950 Mhz. However, some peaks and sudden sharp variation over the higher frequency band of 1 to 5 GHz can be observed. The main reason for such behavior is attributed to the electromagnetic resonance of the soil particles themselves. As the frequency becomes higher the wavelength becomes smaller comparable to the particle's sizes. Moreover, the gray soil has the highest conductivity (around 20 mS/M for dry sample) among the soil samples where its content of iron is higher than others. As a result of the high value of the electrical conductivity, the gray soil has the highest tangential loses which will increase the overall wave propagation losses as will be discussed later. The yellow sand sample has the highest relative permittivity for a dry sample (around 2.9 at 950 MHz). The red sand sample exhibits lower values of electrical permittivity compared with the yellow sand (around 0.5 difference), while other parameters have almost the same behavior. The black soil shows the smoothest variations over the higher frequency band in terms of all measured parameters. This facet can be attributed to the sample composition of fine particulate matter that has electromagnetic resonance out of the observing band. Both white gravels and black soil samples attain the lowest permittivity among the tested samples. The highest conductivity and tangential losses can be noticed in the red/orange gravel sample especially over the lower band, which is a direct result of higher ferrous composition of this sample.

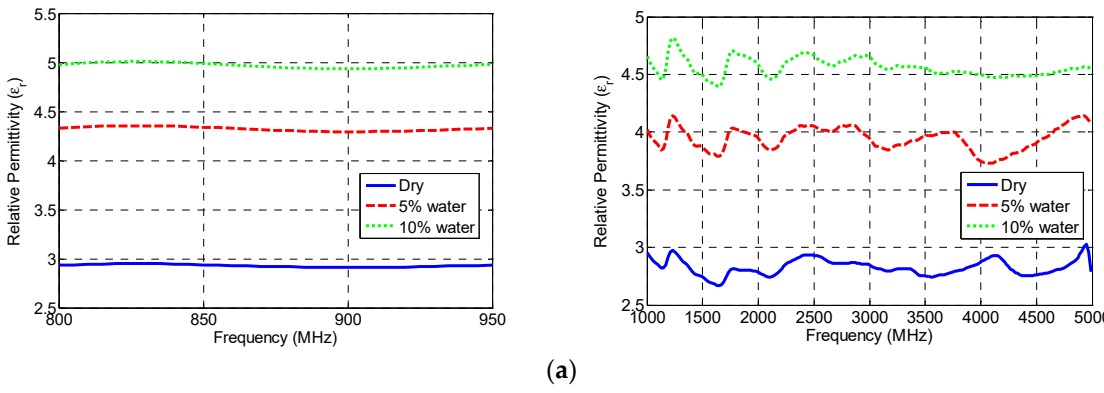

(**a**)

**Figure 2.** *Cont.*

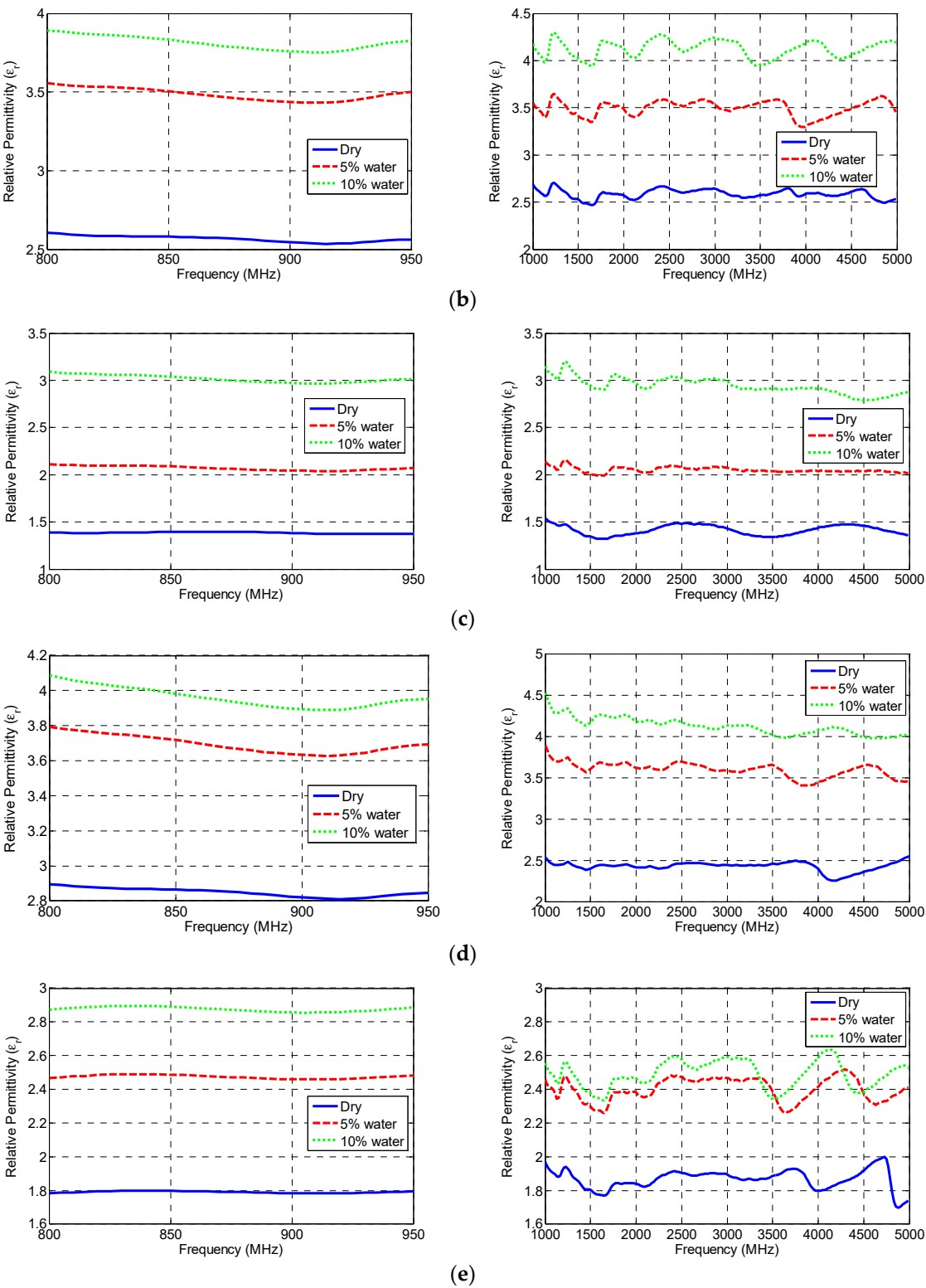

**Figure 2.** *Cont.*

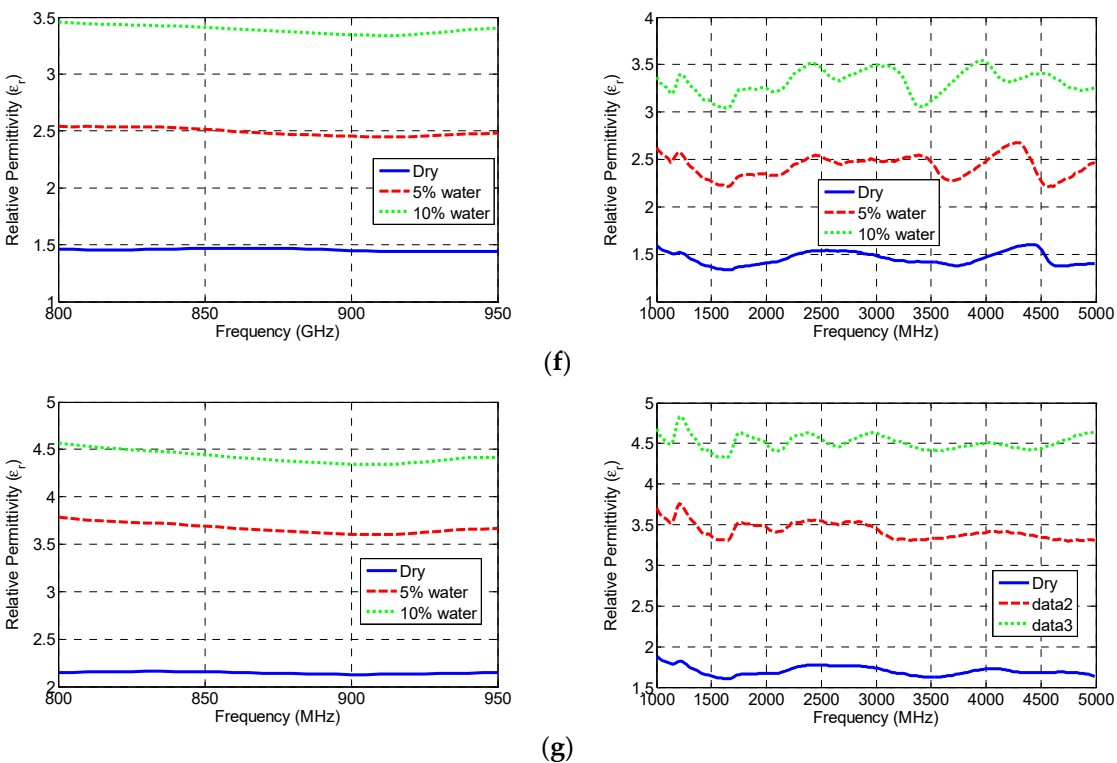

(f)

(g)

**Figure 2.** The Relative Permittivity (real part). (**a**) Yellow sand, (**b**) Red sand, (**c**) Black soil, (**d**) Gray soil, (**e**) Black gravels, (**f**) White gravels, (**g**) Red/Orange gravels.

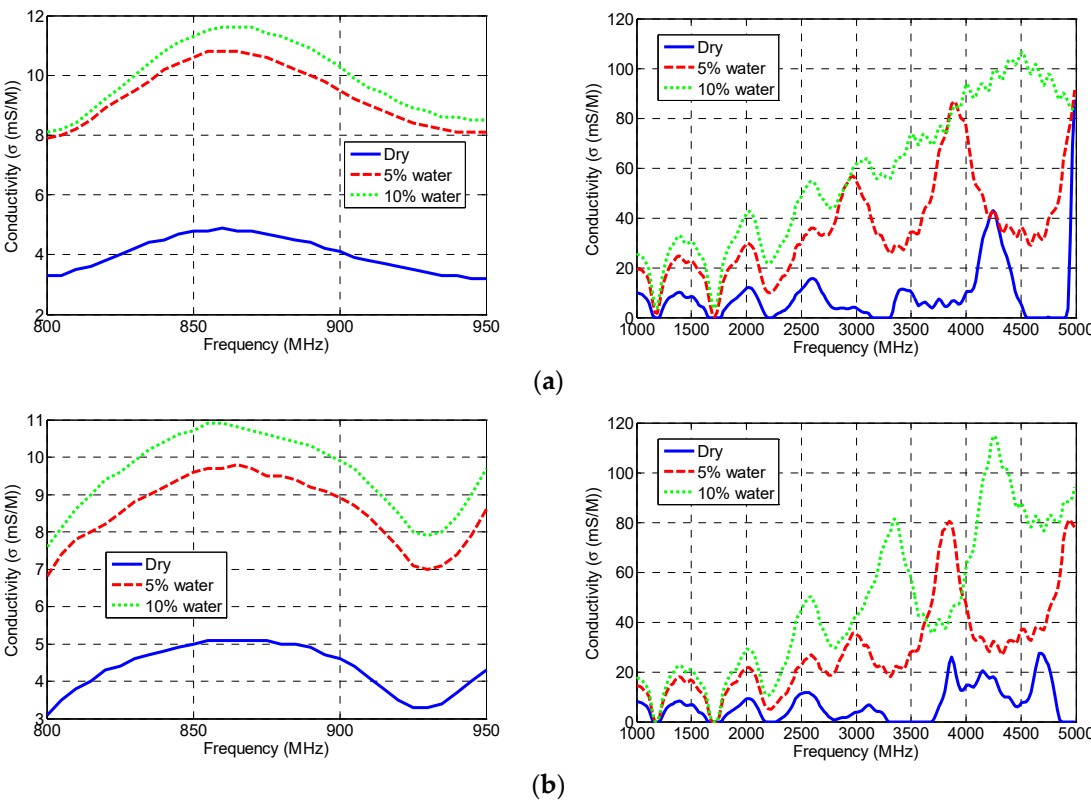

(a)

(b)

**Figure 3.** *Cont.*

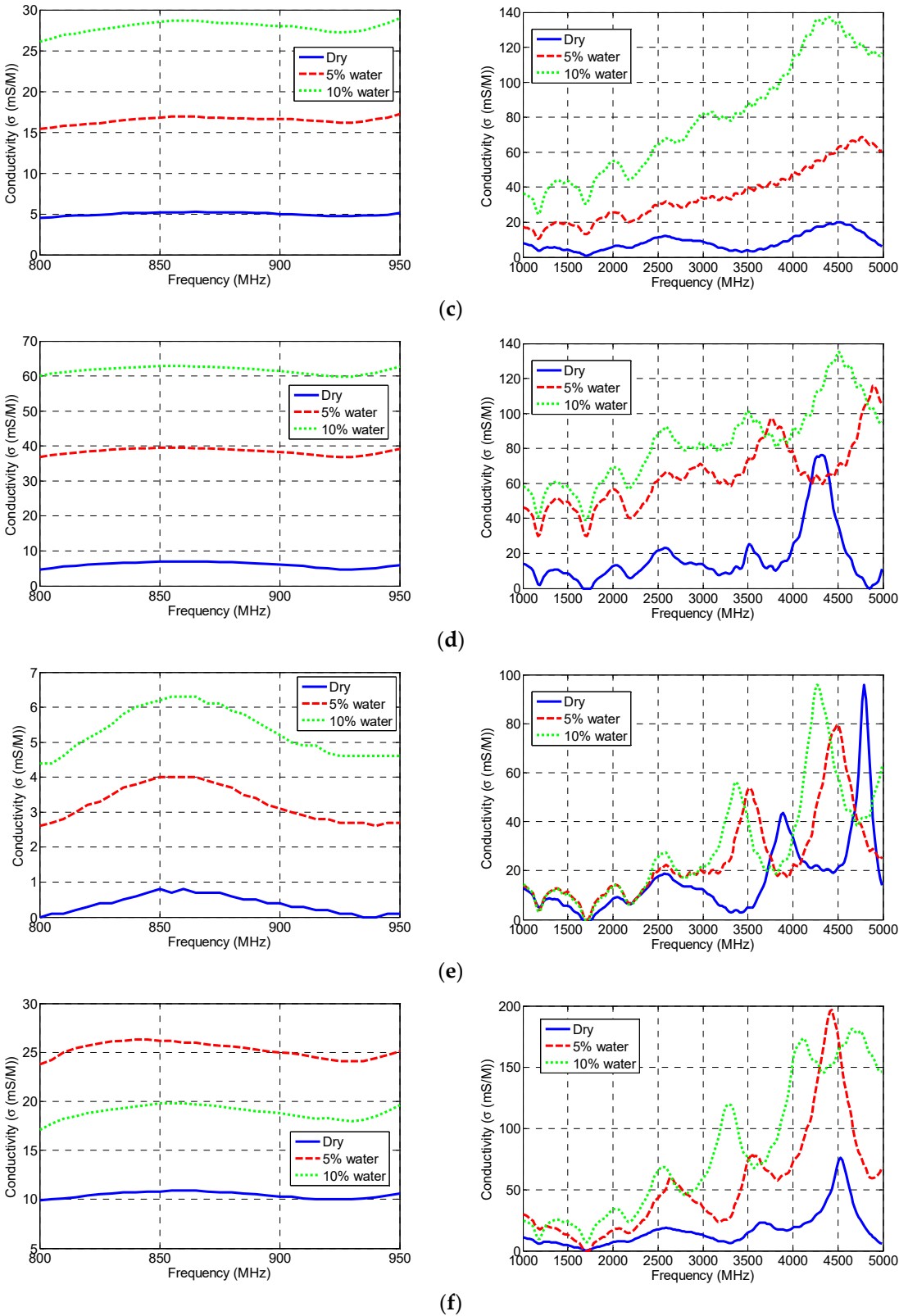

**Figure 3.** *Cont.*

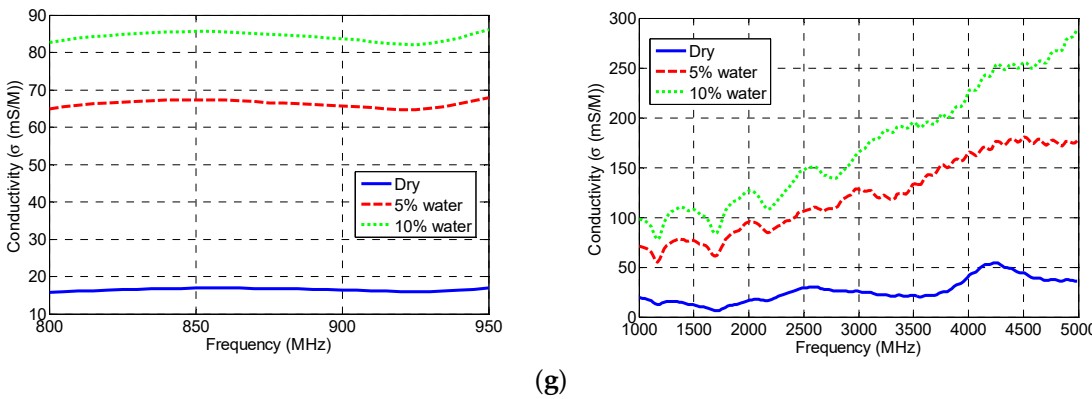

(**g**)

**Figure 3.** The electrical conductivity (mS/M). (**a**) Yellow sand, (**b**) Red sand, (**c**) Black soil, (**d**) Gray soil, (**e**) Black gravels, (**f**) White gravels, (**g**) Red/Orange gravels.

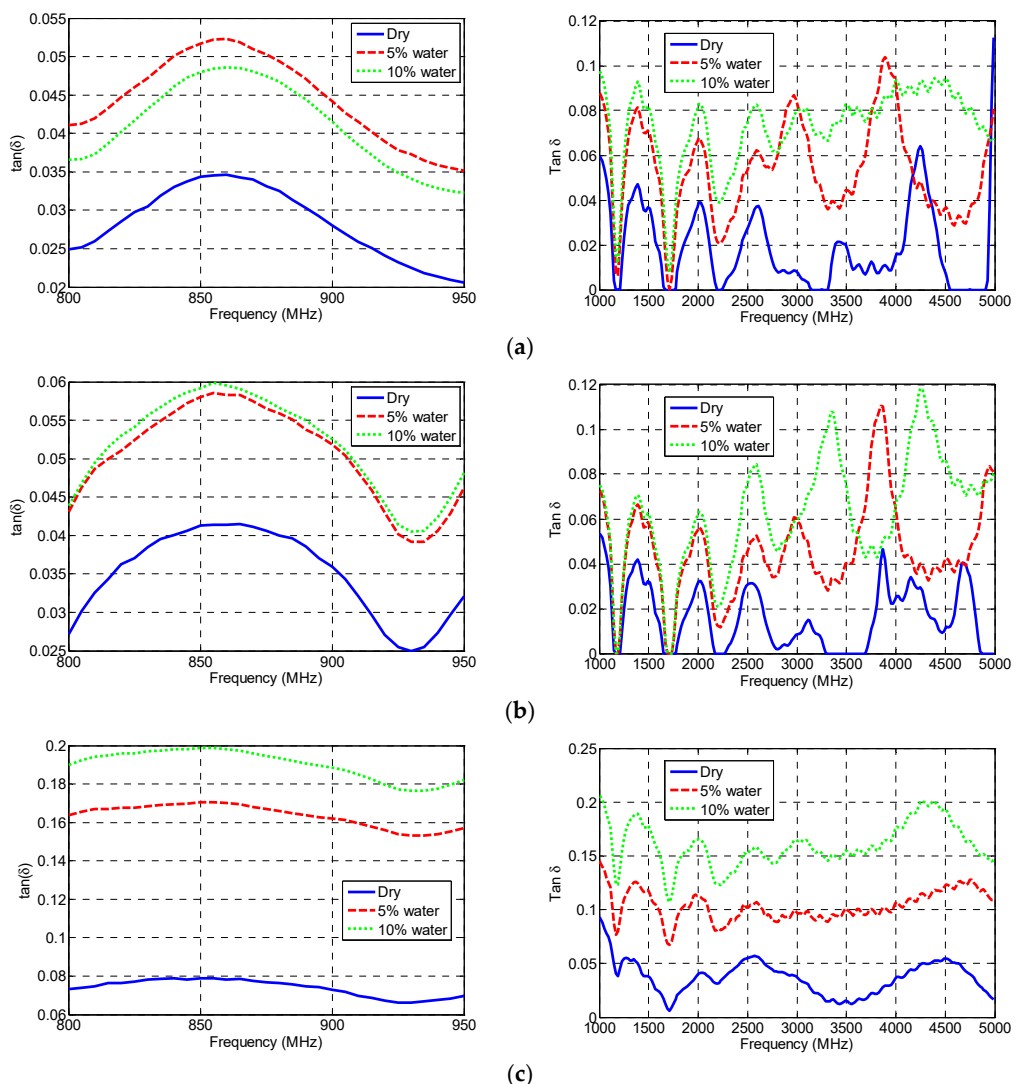

(**a**)

(**b**)

(**c**)

**Figure 4.** *Cont.*

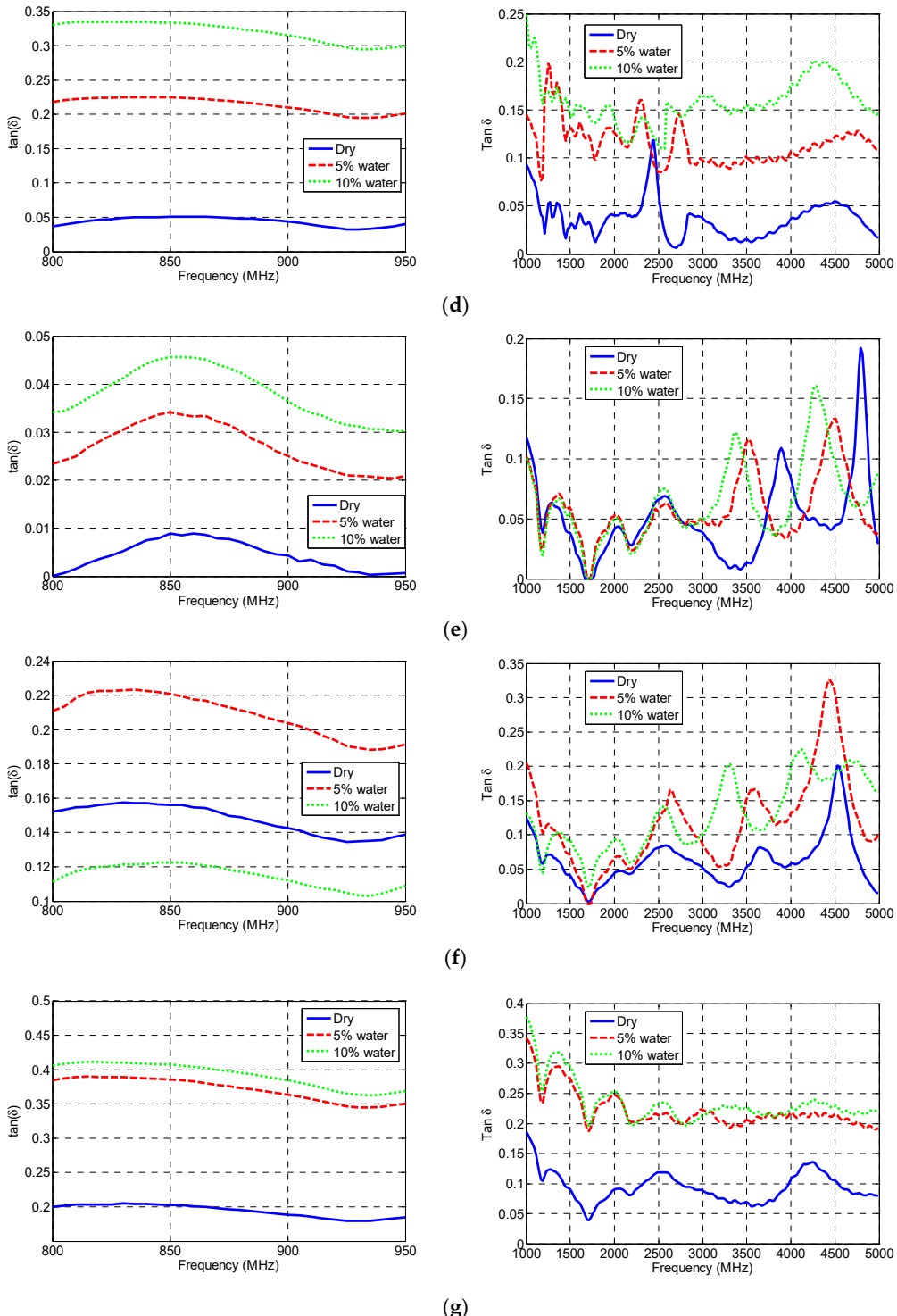

**Figure 4.** The measured tangential losses (*tanδ*). (**a**) Yellow sand, (**b**) Red sand, (**c**) Black soil, (**d**) Gray soil, (**e**) Black gravels, (**f**) White gravels, (**g**) Red/Orange gravels.

As illustrated in Table 1, an abbreviated summary of the obtained results is included. This table can be used in the next step as a fast reference to calculate the total path-loss inside the different soil samples at two frequencies of interest (915 MHz and 2.45 GHz).

**Table 1.** The summarized soil parameters.

| | | 915 MHz | | | 2.45 GHz | |
|---|---|---|---|---|---|---|
| **Yellow Sand** | | | | | | |
| | **Dry** | **5% water** | **10% water** | **Dry** | **5% water** | **10% water** |
| $\varepsilon_r$ *(Real)* | 2.9143 | 4.2973 | 4.942 | 2.9356 | 4.0599 | 4.6842 |
| $\varepsilon_r$ *(Imag.)* | 0.0732 | 0.1728 | 0.184 | 0.0642 | 0.2026 | 0.3269 |
| *Conductivity ($\sigma$ (S/M))* | 0.0037 | 0.0088 | 0.0094 | 0.0088 | 0.0276 | 0.0446 |
| *Tan ($\delta$)* | 0.0251 | 0.0402 | 0.0372 | 0.0219 | 0.0499 | 0.0698 |
| **Red Sand** | | | | | | |
| | **Dry** | **5% water** | **10% water** | **Dry** | **5% water** | **10% water** |
| $\varepsilon_r$ *(Real)* | 2.5346 | 3.4321 | 3.7533 | 2.6658 | 3.5885 | 4.2652 |
| $\varepsilon_r$ *(Imag.)* | 0.0749 | 0.1574 | 0.1752 | 0.0666 | 0.1495 | 0.2909 |
| *Conductivity ($\sigma$ (S/M))* | 0.0038 | 0.008 | 0.0089 | 0.0091 | 0.0204 | 0.0396 |
| *Tan ($\delta$)* | 0.0296 | 0.0459 | 0.0467 | 0.025 | 0.0417 | 0.0682 |
| **Black Soil** | | | | | | |
| | **Dry** | **5% water** | **10% water** | **Dry** | **5% water** | **10% water** |
| $\varepsilon_r$ *(Real)* | 1.3742 | 2.0395 | 2.9666 | 1.4855 | 2.0925 | 3.0339 |
| $\varepsilon_r$ *(Imag.)* | 0.0944 | 0.3221 | 0.5418 | 0.0785 | 0.2141 | 0.4552 |
| *Conductivity ($\sigma$ (S/M))* | 0.0048 | 0.0164 | 0.0276 | 0.0107 | 0.0292 | 0.062 |
| *Tan ($\delta$)* | 0.0687 | 0.1579 | 0.1826 | 0.0528 | 0.1023 | 0.15 |
| **Gray Soil** | | | | | | |
| | **Dry** | **5% water** | **10% water** | **Dry** | **5% water** | **10% water** |
| $\varepsilon_r$ *(Real)* | 2.8096 | 3.6294 | 3.8903 | 2.456 | 3.6897 | 4.1938 |
| $\varepsilon_r$ *(Imag.)* | 0.1042 | 0.7353 | 1.1863 | 0.0427 | 0.3896 | 0.6397 |
| *Conductivity ($\sigma$ (S/M))* | 0.0053 | 0.0374 | 0.0604 | 0.0058 | 0.0531 | 0.0872 |
| *Tan ($\delta$)* | 0.0371 | 0.2026 | 0.3049 | 0.0174 | 0.1056 | 0.1525 |
| **Black Gravels** | | | | | | |
| | **Dry** | **5% water** | **10% water** | **Dry** | **5% water** | **10% water** |
| $\varepsilon_r$ *(Real)* | 1.7835 | 2.4591 | 2.8563 | 1.9114 | 2.4819 | 2.5975 |
| $\varepsilon_r$ *(Imag.)* | 0.0044 | 0.0554 | 0.0953 | 0.1176 | 0.1384 | 0.1665 |
| *Conductivity ($\sigma$ (S/M))* | 0.0002 | 0.0028 | 0.0049 | 0.016 | 0.0189 | 0.0227 |
| *Tan ($\delta$)* | 0.0025 | 0.0225 | 0.0334 | 0.0615 | 0.0558 | 0.064 |
| **White Gravels** | | | | | | |
| | **Dry** | **5% water** | **10% water** | **Dry** | **5% water** | **10% water** |
| $\varepsilon_r$ *(Real)* | 1.4404 | 2.4468 | 3.3394 | 1.5407 | 2.5438 | 3.5044 |
| $\varepsilon_r$ *(Imag.)* | 0.197 | 0.4809 | 0.3582 | 0.1202 | 0.2842 | 0.4292 |
| *Conductivity ($\sigma$ (S/M))* | 0.01 | 0.0245 | 0.0182 | 0.0164 | 0.0387 | 0.0585 |
| *Tan ($\delta$)* | 0.1368 | 0.1965 | 0.1073 | 0.078 | 0.1117 | 0.1225 |
| **Red Orange Gravels** | | | | | | |
| | **Dry** | **5% water** | **10% water** | **Dry** | **5% water** | **10% water** |
| $\varepsilon_r$ *(Real)* | 1.7257 | 3.6068 | 4.3447 | 1.7772 | 3.5538 | 4.5955 |
| $\varepsilon_r$ *(Imag.)* | 0.3158 | 1.2755 | 1.6212 | 0.2042 | 0.764 | 1.0555 |
| *Conductivity ($\sigma$ (S/M))* | 0.0161 | 0.0649 | 0.0825 | 0.0278 | 0.1041 | 0.1439 |
| *Tan ($\delta$)* | 0.183 | 0.3536 | 0.3731 | 0.1149 | 0.215 | 0.2297 |

## 3. Link Budget Analysis of the Soil

Now, the collected data in the previous section are used to arrive at an approximated link budget. The obtained link budget model can be employed to calculate the total losses due to the wave propagation inside the soil. To achieve this goal, a MATLAB code is developed assuming the following parameters:

- Reader antenna gain of 7 dBi
- Tag antenna gain of 2 dBi
- Total radiated power of the reader (*Pt*) is 30 dBm
- Both reader and tag antennas are −20 dB matched
- The first order reflection from the soil layer is considered
- Full homogenous soil material

The attenuation coefficient due to wave propagation inside the soil $\alpha$ can be estimated using Equation (1), while the total received power $P_r$ can be obtained using Equation (2)

$$\gamma = \alpha + j\beta = j\omega\sqrt{\mu\epsilon}\sqrt{1 - j\frac{\sigma}{\omega\epsilon}} \tag{1}$$

$$\frac{P_r}{P_t} = G_t G_r \left(\frac{\lambda}{4\pi R}\right)^2 \left(1 - |\Gamma_t|^2\right)\left(1 - |\Gamma_r|^2\right)e^{-\alpha R} \tag{2}$$

where $\gamma$ is the propagation constant, $\beta$ is the phase constant, $\omega$ is the radial frequency, $\sigma$ is the electrical conductivity, $\epsilon$ is the soil permittivity, $G_t$ and $G_r$ are reader antenna and the tag antenna gain, respectively, $\Gamma_t$ and $\Gamma_r$ $r$ are the reader antenna and the tag antenna reflection coefficient, respectively and $R$ is the soil thickness in meters.

Figure 5 shows the calculated link budget analysis graphs for different soil types (dry soil) over two operating bands, from 850 to 950 MHz, and from 1 to 5 GHz. The graphs are plotted for different soil thicknesses ranged from 0.5 m up to 4 m. A smooth variation in the total path-loss in the lower band can be noticed, while there are some up-rising peaks and fast variations in the second band. This phenomenon can be attributed to the soil inhomogeneity and the presence of some big stones compared with the wavelength which cause this kind of discrepancy.

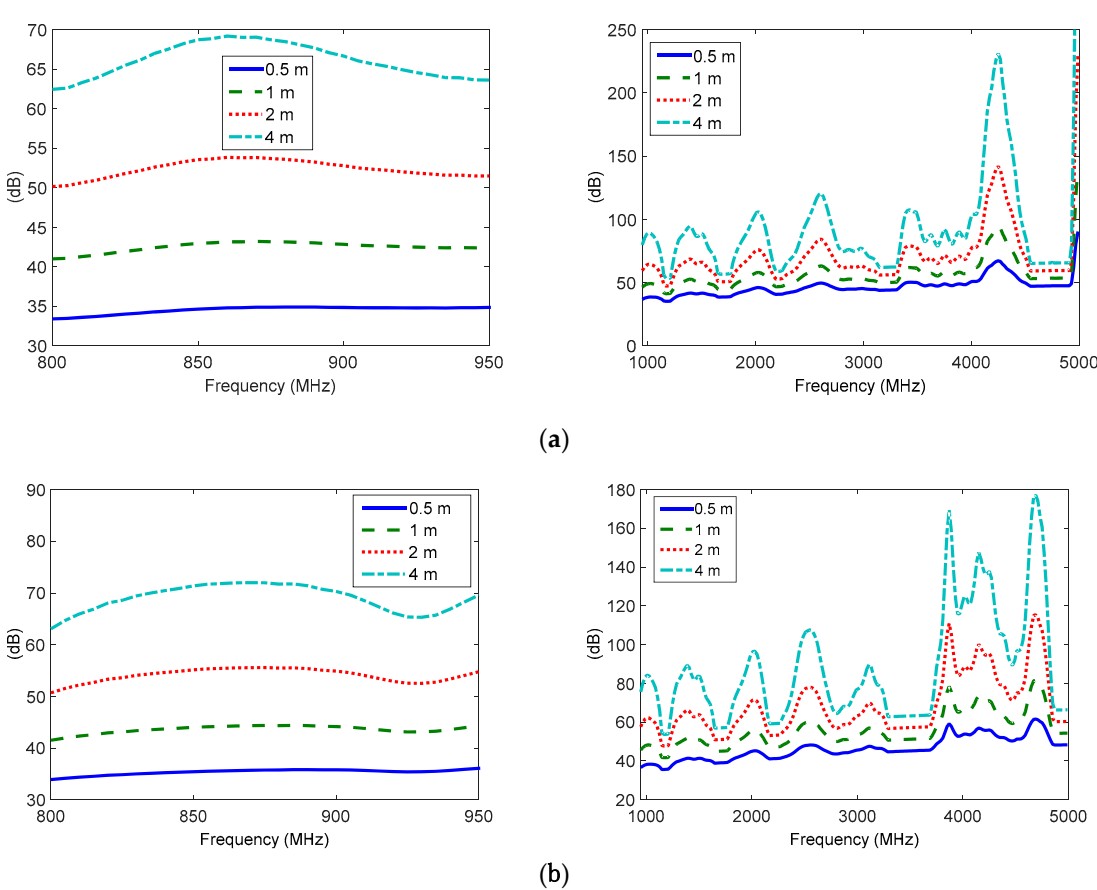

(a)

(b)

**Figure 5.** *Cont.*

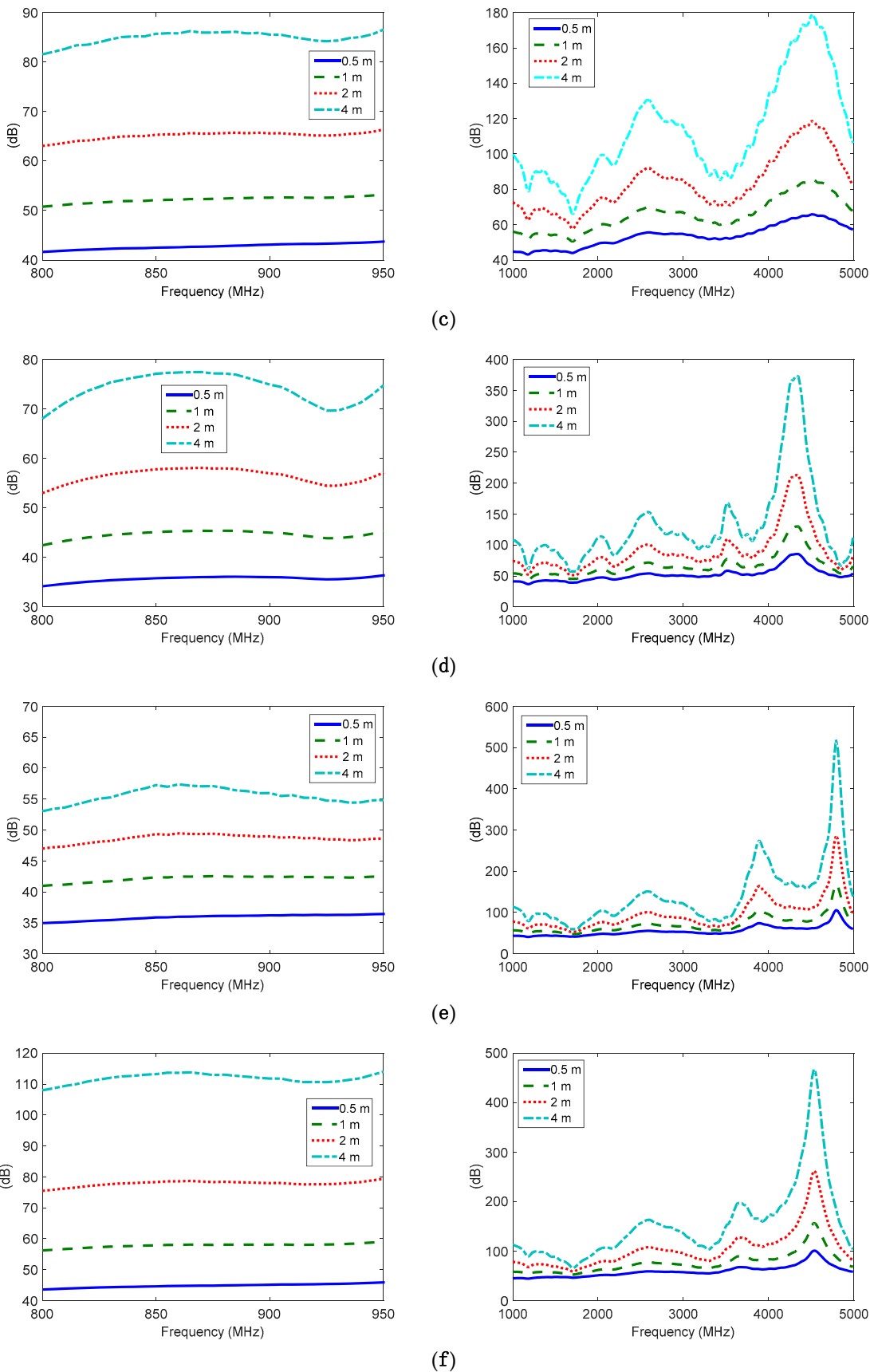

**Figure 5.** *Cont.*

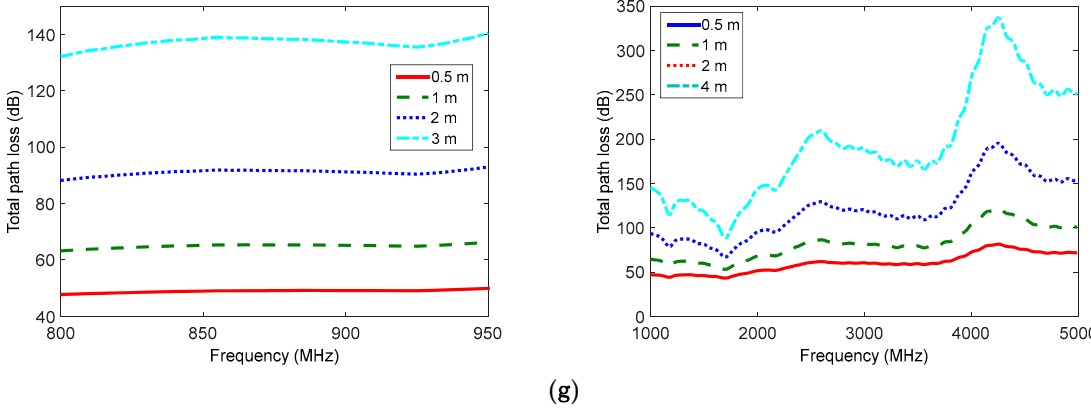

(g)

**Figure 5.** The calculated link budget analysis of different soil types. (**a**) Yellow sand, (**b**) Red sand, (**c**) Black soil, (**d**) Gray soil, (**e**) Black gravels, (**f**) White gravels, (**g**) Red/Orange gravels.

## 4. Practical Verification of the Link Budget Model Using a Buried Metal-Backed RFID Tag Antenna

In order to test the validity of the link budget analysis model, a simple practical test for a buried tag localization and data reading is made. In this experiment we used the tag antenna proposed in [15]. The tag antenna was fabricated using photolithographic method. This tag was designed to be backed by a metallic surface, as it is placed on the pipe surface. A photo for the fabricated tag antenna (before RFID chip mounting) is illustrated in Figure 6.

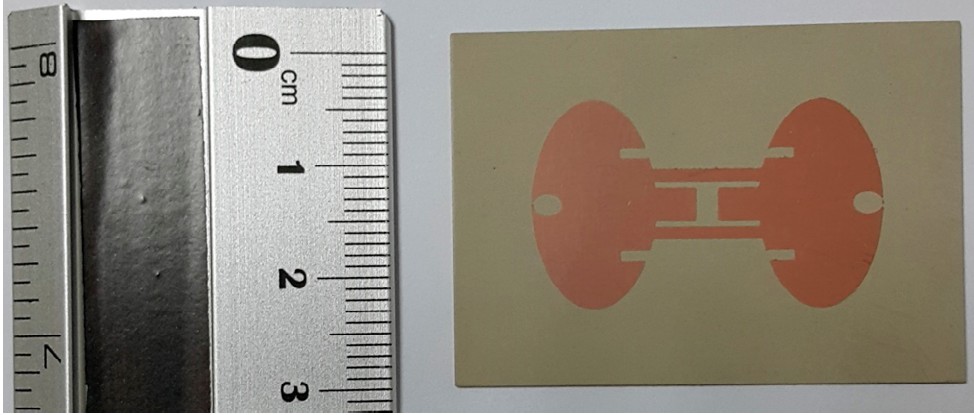

**Figure 6.** The fabricated metal backed Radio Frequency Identification (RFID) tag antenna.

The tag antenna port is a differential port and it is intended to be connected to MONZA3 chip [16], which has an input impedance of (32-j216) ohm. The proposed tag should have impedance close to the complex conjugate of the chip impedance. To measure practically the input impedance of the antenna, an imaging theory procedure explained in [17] is adopted. In this theory, the dipole antenna is divided into two symmetric parts, where each one of them is equivalent to a monopole antenna. This half is placed over a finite ground plane made of copper, and connected to the VNA to measure its input impedance as indicated in Figure 7. The main objective of this method is to use the unbalanced monopole antenna to create a balanced dipole antenna by this ground plane. The input impedance of the dipole can be evaluated from:

$$Z_{\text{Dipole}} = 2 \times Z_{\text{monopole}} \qquad (3)$$

The input impedance of the fabricated tag antenna is shown in Figure 8. The antenna reflection coefficient is calculated using the formula stated in [15]. The measured reflection coefficient of the

proposed antenna shows very good performance around the operating frequency of 915 MHz. Figure 9 presents the reflection coefficient S$_{11}$ of the proposed tag antenna.

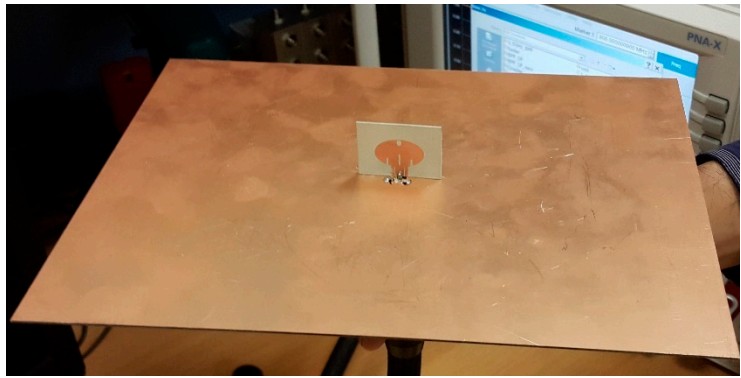

**Figure 7.** Practical measurement of the input impedance of the proposed tag antenna.

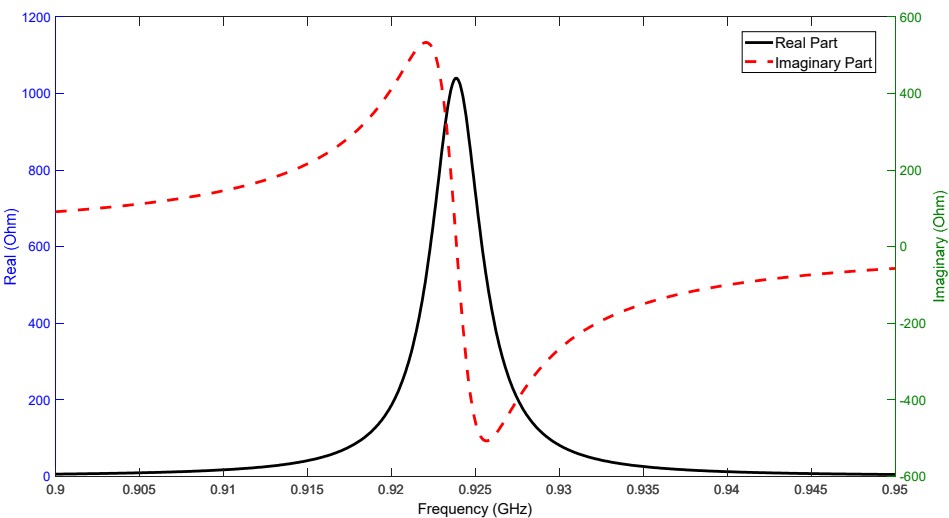

**Figure 8.** The input impedance of the proposed tag antenna (real and imaginary).

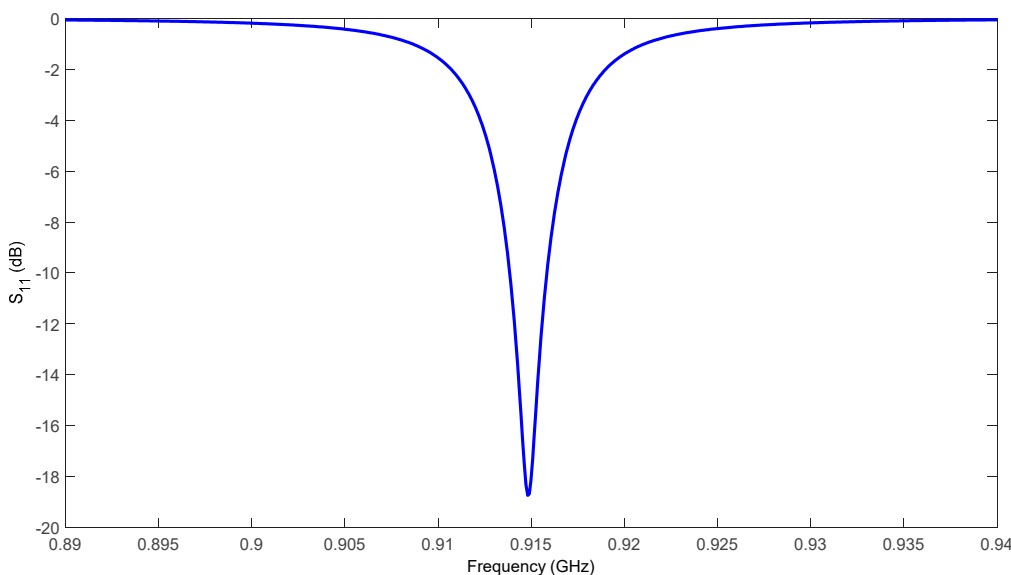

**Figure 9.** The reflection coefficient S$_{11}$ of the proposed tag antenna.

To perform the practical experiment, the tag antenna with the RFID chip is buried under a 40 cm layer of yellow sand contained in the center of a plastic container as shown in Figure 10. The plastic container has the dimensions of 60 cm × 40 cm × 50 cm. For tag localization and data collecting, we built the system presented in Figure 11. This measurement system is based on the Impinj speedway R420 system for RFID reading [18]. This system, shown in Figure 11, is connected to a log periodic antenna with a gain of 7 dBi at 915 MHz to operate as a reading antenna. The RFID reading system is connected to a PC through a network router. Finding an exact location is easier than ever before. The reader transmits a signal to the buried tag and the tag returns a signal to the reader. The position is confirmed, even through the toughest soil conditions. The soil sample is assumed to be dry, and the experiment is done at room temperature.

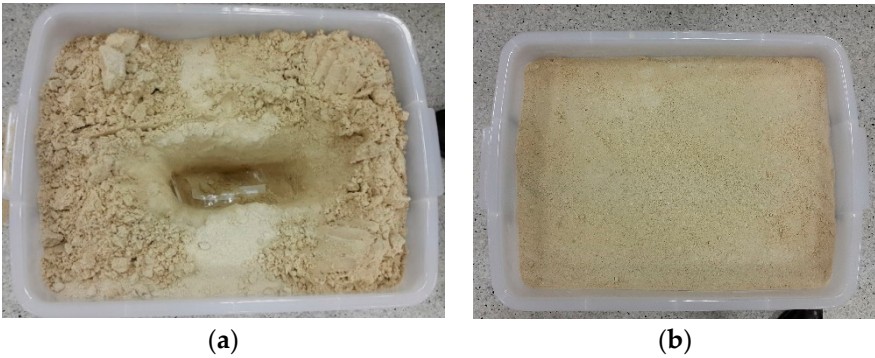

| (a) | (b) |

**Figure 10.** The sand tank used. (**a**) before burying the tag. (**b**) after burying the tag.

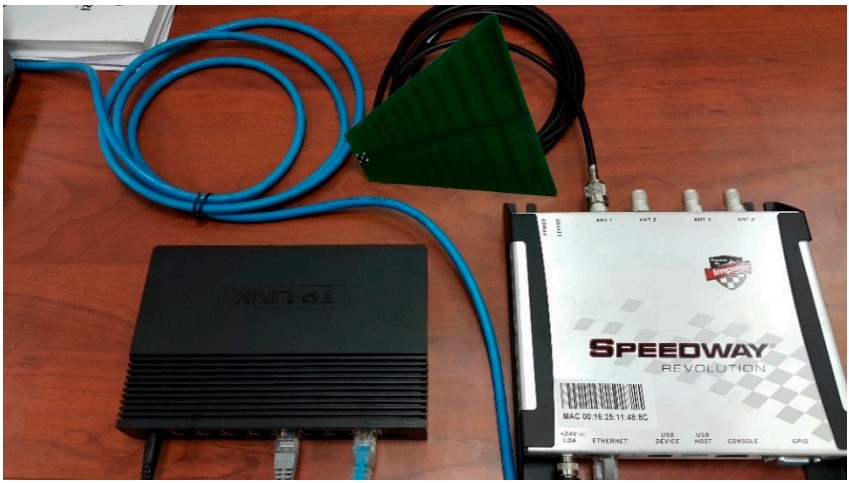

**Figure 11.** The RFID reader system setup.

The Impinj tag reading system R420 can provide a transmitted power up to 30 dBm. The reading antenna is mounted above the sand container at a certain distance *R*. The transmitted power travels from the reading antenna through the air then penetrates the soil to hit and activate the buried RFID tag. The tag starts to send the information back to reading system through both soil then air. The expected received power at the reading system can be calculated as:

Expected received power = Tx power (in dBm) − 2 × [soil attenuation (in dB) + FSPL (in dB)] (4)

where FSPL is the Free Space Power Loss (in dB) and it can be calculated from [19]:

$$FSPL = 20 \log R + 20 \, Log \, f - 27.55 \tag{5}$$

where $R$ is the reading distance from the sand tank surface to the reading antenna (in m), and $f$ is the frequency in Mhz.

Factor 2 in Equation (3) is inserted to compensate the round trip of the electromagnetic wave (from the reading antenna to the tag then from the tag to the reading antenna). Now for the yellow sand case, according to Equation (2) an attenuation of 32 dB can be evaluated for a 40 cm thickness of the soil.

When the distance $R$ is set to be 0.5 m above the sand box, tag localization and its stored data can be fetched successfully. The expected received power from the buried tag is expected to be −71.3 dB, however the actual received power is found to be −73 dB.

The discrepancy between the expected and the actual received power (3.85% error percent) can be attributed to the multipath effect form of the surrounding objects and from the edges of the container. These undesired reflections act as new illumination sources for the tag antenna and result in higher received signals than what is expected from the total path loss calculations. Fortunately, this situation cannot occur over wide areas of land, and the accuracy is still within an acceptable margin.

## 5. Conclusions

In this work, a detailed EM soil characterization for different seven soil samples is presented. Relative permittivity, electrical conductivity, and tangential losses are estimated at room temperature for the soil samples. The obtained results were employed to conduct a complete link budget analysis study for EM wave propagation inside the soil. The humidity effect was taken into consideration and 5% and 10% water content were considered in the modeling process. For link budget modeling verification, an RFID tag was buried under 40 cm of yellow sand. The tag antenna was successfully read at 0.5 m above the surface of the sand.

**Author Contributions:** The research methodology in the manuscript is developed by A.E., K.J. and A.S. The practical experiments are conducted by A.E. The manuscript is written by A.E. and revised by A.T. and A.S. All of the work presented in this manuscript is done under supervision of A.T., K.J. and A.S. All authors have read and agreed to the published version of the manuscript.

**Funding:** This Project was funded by the National Plan for Science, Technology and Innovation (MAARIFAH), King Abdulaziz City for Science and Technology, Kingdom of Saudi Arabia, Award Number (12-ELE2462-02).

**Conflicts of Interest:** The authors declare no conflict of interest.

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
