# Peer review of "Electromagnetic Soil Characterization for Undergrounded RFID System Implementation"

_electronics, doi:10.3390/electronics9010106_

Round 1

Reviewer 1 Report

I observed much improvement in the paper. However, I have still following concerns. Therefore, I recommended major revision.

1. The discussion on the measurement result was added. Some of the dicsussion, for example iron contained in gray soil, are interesting. However, the discussion is not comorehensive; only the black gravels, gray soil, yellow sand are mentioned in the added paragraph. Can you mention other samples?

2. The author added the expected received power in Table II to compare the link-budget analysis and the measurement, which is an important result. However, a big discrepancy was observed. The author explained that the reason is multipath. Regarding this, my concerns are as follows.

2.1 There is no evidence that the multipath is the reason.

2.2 If so, this indicates that the experiment condition was not well designed.

Author Response

Thank you for your comments. We have carefully reviewed the comments and have revised the manuscript accordingly. Our responses are given in a point-by-point manner below. Changes to the manuscript are shown in highlighted green font. We hope the revised version is now suitable for publication.

Detailed Reply

I observed much improvement in the paper. However, I have still following concerns. Therefore, I recommended major revision.

The discussion on the measurement result was added. Some of the dicsussion, for example iron contained in gray soil, are interesting. However, the discussion is not comorehensive; only the black gravels, gray soil, yellow sand are mentioned in the added paragraph. Can you mention other samples?

A new paragraph is added to the paper. The paragraph illustrates some observations regarding to the other samples (black soil, white gravels, Red sand, and the Red/Orange gravels).

Please check page 3 line 114 to 122.

The new paragraph is “The red sand sample exhibits lower values of electrical permittivity compared with the yellow sand (around 0.5 difference), while other parameters have almost the same behavior. The black soil shows the smoothest variations over the higher frequency band in terms of all measured parameters. This attitude can be attributed to the sample composition of fine particulate matter that has electromagnetic resonance out of the observing band. Both white gravels and black soil samples attain the lowest permittivity among the tested samples. The highest conductivity and tangential losses can be noticed in red / orange gravel sample especially over the lower band, which is a direct result of higher ferrous composition of this sample”.

The author added the expected received power in Table II to compare the link-budget analysis and the measurement, which is an important result. However, a big discrepancy was observed. The author explained that the reason is multipath. Regarding this, my concerns are as follows.

2.1 There is no evidence that the multipath is the reason.

Actually, there is an evidence for this phenomenon. If we look to the error in the first case (i.e. 0.5 m above the sand tank) the difference between the expected power and the measured power is about 2.3 dB difference (around 3.15%). When the distance increases, the reader antenna main beam becomes wider and starts to hit the edges of the tank and some surrounding objects casing multi-source illumination to the RFID tag and the error percent starts to increase (around 11.2% error is achieved when the distance is around 1 m). Increasing the distance more increases the multipath (the error percent when the distance becomes 2.5m is 12.65 %). In abbreviation, the discrepancy between the expected power and the measured power increases with distance increase (i.e. increasing the multipath phenomenon).

2.2 If so, this indicates that the experiment condition was not well designed.

In real life, localizing and reading the buried RFID tag are much easier. No need at all to rise the reader antenna above the soil more than 0.5m (sometimes it is placed above much smaller distance). This scenario, up on the experiment observations, shows that the measured signal will not be affected much by any surrounding objects.

What we do in the experiment is pushing the limits to increase the attenuation in the transmitted signal as much as we can. We had two ways to do that even increase the sand height more than 40 cm or to increase the reading distance above the sand box. The first solution was not applicable as the weight of the sand box becomes too heavy to carry (more than 60 KG). The second solution is to increase the reading distance which is adopted by our team. This solution results in some errors (12.65 for the worst case), however the errors can be tolerated and the experiment still gives good indications about the amount of received power.

Reviewer 2 Report

It was confirmed that the re-submitted paper was carefully corrected by the authors.

Author Response

Thank you very much.

Reviewer 3 Report

You present an interesting soil characterization method using passive RFID tags. You've presented multiples tests, however, the results occupy most of the available space. Please, merge plots and tables in a more efficient way to improve the aspect.

In addition, how is this process better than other procedures? Not everyone has a network analyzer (it costs more than 2000 dollars). Is it possible to get similar results using a regular RFID reader?

You must improve the experiment explanation and results.

Author Response

Thank you for your comments. Our responses are given in a point-by-point manner below. Changes to the manuscript are shown in highlighted green font. We hope the revised version is now suitable for publication.

Detailed Reply

You present an interesting soil characterization method using passive RFID tags.

We did not present soil characterization method using passive RFID tags in this paper.

Actually, the paper consists of two main parts, the first one presents EM soil characterization using The Dielectric Assessment Kit (DAK) provided by SPEAG. This kit consists of a measuring probe, cable, and supporting holder. The probe is calibrated and connected to a vector network analyzer. The outcome of this experiment are the different soil parameters including permittivity, conductivity, and tangential losses. The obtained parameters are used to come out with a link budget model illustrates the possible attenuation inside the soil. This model can be used for many applications such as underground RFID and Ground Penetrating Radar (GPR).

The second part of the paper discusses a practical verification of the obtained link budget model using a buried metal-backed passive RFID tag antenna under 40 cm sand layer. The attenuation model is used to estimate the round-trip losses inside the soil. Then based on these calculations we can predict the power received from the RFID tag antenna and compare it with the actual measured power at different heights. The power measurement is carried out using Impinj Speedway R420 RFID reader (Not the VNA).

You've presented multiples tests, however, the results occupy most of the available space. Please, merge plots and tables in a more efficient way to improve the aspect.

I agree with you, the paper presented many figures and plots, however I think it is mandatory to illustrate the parameters variations over the measurement bands. The paper presents three different parameters for seven soil samples for three different humidity scenarios over two different frequency bands.

For example, if we merge the two plots of the two frequency bands in just one plot, i.e. making one plot extends from 800 MHz to 5000 MHz, the results of doing that is losing a lot of details and minor variations especially at the lower band (from 800 MHz to 950 MHz). Another example, if we merge the same parameter, (like the permittivity), for all the soil samples in one plot, the result will be a dense graph which is difficult to read. Merging curves is an easy task, however important data will be lost.

In addition, how is this process better than other procedures?

Using DAK kit and VNA for soil characterization provide a faster process to measure multiple EM parameters including (real and complex permittivity, electrical conductivity, tangential losses). The accuracy of this type of measurements is very good compared with other time consuming, more expensive, and more complicated methods used to figure out the soil parameters. We discussed other soil characterization methods in the introduction section (page 2 line 45 to 60).

Not everyone has a network analyzer (it costs more than 2000 dollars).

I totally agree with you that not every one can buy a VNA as it costs more than 70,000 USD not just 2000 USD. For this reason, we wrote this paper to be a reference for the researchers to provide them with detailed soil parameters at important frequency bands with no need to have a VNA nor an EM probe.

Is it possible to get similar results using a regular RFID reader?

No, regular RFID readers can not be used for soil characterization, i.e., it can afford EM results such as permittivity, conductivity, and tangential losses. This kind of RFID reader can be used for estimating the received power from the RFID tag antenna only.

You must improve the experiment explanation and results.

A new paragraph is added to the paper. The paragraph illustrates some observations regarding to the other samples (black soil, white gravels, Red sand, and the Red/Orange gravels).

Please check page 3 line 114 to 122.

The new paragraph is “The red sand sample exhibits lower values of electrical permittivity compared with the yellow sand (around 0.5 difference), while other parameters have almost the same behavior. The black soil shows the smoothest variations over the higher frequency band in terms of all measured parameters. This attitude can be attributed to the sample composition of fine particulate matter that has electromagnetic resonance out of the observing band. Both white gravels and black soil samples attain the lowest permittivity among the tested samples. The highest conductivity and tangential losses can be noticed in red / orange gravel sample especially over the lower band, which is a direct result of higher ferrous composition of this sample”.

Round 2

Reviewer 1 Report

The added discussion about the soil characterization was sufficient. Then, my only remaining concern is the result in Section 4, especially Table 2, and the authors' explanation. Because the result and experimental setup were not justified sufficiently, I had to recommend major revision again. Please refer to the following comments.

Regarding to the following reply,

> In real life, localizing and reading the buried RFID tag are much easier. No need at all to rise the reader antenna above the soil more than 0.5m (sometimes it is placed above much smaller distance).

I am afraid that your explanation means that the experiment was not well designed. The experiment should be closer to real scenario as much as possible.

Regarding to the following reply,

> What we do in the experiment is pushing the limits to increase the attenuation in the transmitted signal as much as we can.

I am afraid there is no scientific meaning of "putshing the limits to increase the attenuation" from the view point of the focus of this paper. The focus of this paper is soil characterization and link-budget analysis, not  free space propagation. Condiering this, if the attenutation is increased by a soil thickness, this increase will have a meaning. The attenuation increased by free-space propagation does not have meaninig. 

Regarding to the following reply,

This solution results in some errors (12.65 for the worst case), however the errors can be tolerated and the experiment still gives good indications about the amount of received power.

This might be subjective, but I don't think it is an acceptable error. Personally, 1 to 3 dB is acceptable as a good agreement; 4 dB to 6 dB depends on the case. 10 dB might be too big in this case.

Author Response

Thank you very much for your valuable comments.

The manuscript is edited according to your comments and Table 2 is deleted. only one case is considered (when the distance R= 0.5 m). The last two paragraphs in page 14 are edited.

Again Thank you very much. 

Round 3

Reviewer 1 Report

I confirmed that the manuscript was revised and addressed my concern.

This manuscript is a resubmission of an earlier submission. The following is a list of the peer review reports and author responses from that submission.

Round 1

Reviewer 1 Report

Please give more detailed explanation about Figure 1.

Please provide a discussion of the results obtained in the experiment in Chapter 2.

In many cases of the experimental results shown in Figure 2-4, the difference between the 950 MHz and 1000 MHz results is large, and the characteristics between these two frequencies appear to be discontinuous.
Please explain why this shift occurs.

Gamma in the text of the third paragraph of Chapter 3 is not displayed.

The same contents as the third and fourth paragraphs in Chapter 3 appear in the fifth and sixth paragraphs.

Is the content of Chapter 4 complete?

The picture in Fig. 10 cannot fully confirm the situation where the tag is placed.
Please consider a more appropriate expression.

Is it only the case of yellow sand that has evaluated the communication performance of tags buried underground?
Did the difference in soil moisture affect communication performance?

Please define the communication distance in Table 2.

It is not clear in what situation the experimental results shown in Chapter 4 were obtained. Please add a more detailed description.

Is a conclusion chapter unnecessary for this paper?

Reviewer 2 Report

I agree that soil EM characterization and link-budget analysis are important research topic. However, from the following observations, I had to recommend rejection of this paper.

1. In the introduction, literature on soil EM characterization should be reviewed in detail to show the novelty of this paper. Currently, only [8] is refered, just mentioning that measurement method is different. This might be too weak to support the novelty of this paper. I expect, just for example, that the measuerd frequency, types of soil, or findings are new. 

2. Discussion, interpretation, or characterization on the result of the soild EM measurement are needed as a journal paper. Currently, only plots and tables are given without discussion. They are not enough as "characterization". Then, the paper failed to provide novel findings. Also, comparing the current measurement with result in the literature might be needed to justify that the measurement is succesful or to derive new findings.

3. I think the measurement result on the received reading power should be quantitatively compared with the link-budget analysis as a verification.